# Massive methane fluxing from magma–sediment interaction in the end-Triassic Central Atlantic Magmatic Province

Manfredo Capriolo [1,2 ✉], Andrea Marzoli [3 ✉], László E. Aradi [4], Michael R. Ackerson [5], Omar Bartoli [1], Sara Callegaro[2], Jacopo Dal Corso [6], Marcia Ernesto[7], Eleonora M. Gouvêa Vasconcellos [8], Angelo De Min[9], Robert J. Newton [10] & Csaba Szabó[4]

Exceptional magmatic events coincided with the largest mass extinctions throughout Earth's history. Extensive degassing from organic-rich sediments intruded by magmas is a possible driver of the catastrophic environmental changes, which triggered the biotic crises. One of Earth's largest magmatic events is represented by the Central Atlantic Magmatic Province, which was synchronous with the end-Triassic mass extinction. Here, we show direct evidence for the presence in basaltic magmas of methane, generated or remobilized from the host sedimentary sequence during the emplacement of this Large Igneous Province. Abundant methane-rich fluid inclusions were entrapped within quartz at the end of magmatic crystallization in voluminous (about $1.0 \times 10^6 \, km^3$) intrusions in Brazilian Amazonia, indicating a massive (about $7.2 \times 10^3 \, Gt$) fluxing of methane. These micrometre-sized imperfections in quartz crystals attest an extensive release of methane from magma–sediment interaction, which likely contributed to the global climate changes responsible for the end-Triassic mass extinction.

[1] Department of Geosciences, University of Padova, Padova, Italy. [2] Centre for Earth Evolution and Dynamics, University of Oslo, Oslo, Norway. [3] Department of Territory and Agro-Forestry Systems, University of Padova, Legnaro, Italy. [4] Lithosphere Fluid Research Lab, Research and Industrial Relations Center, Faculty of Science, Eötvös Loránd University, Budapest, Hungary. [5] Department of Mineral Sciences, National Museum of Natural History, Smithsonian Institution, Washington DC, USA. [6] State Key Laboratory of Biogeology and Environmental Geology, China University of Geosciences, Wuhan, China. [7] Department of Geophysics, Institute of Astronomy, Geophysics and Atmospheric Sciences, University of São Paulo, São Paulo, Brazil. [8] Geology Postgraduate Program of the Federal University of Paraná, Curitiba, Brazil. [9] Department of Mathematics and Geosciences, University of Trieste, Trieste, Italy. [10] School of Earth and Environment, University of Leeds, Leeds, UK. ✉email: manfredo.capriolo@phd.unipd.it; andrea.marzoli@unipd.it

The geological record shows that massive and rapid inputs of greenhouse gases, such as $CO_2$ and $CH_4$, into the atmosphere–hydrosphere system caused sudden increases of Earth's surface temperature, leading to major environmental changes and severe biotic crises[1–3]. The largest mass extinction events, such as those at the Permian–Triassic boundary and at the end-Triassic, were synchronous with rapid and voluminous magmatism generating Large Igneous Provinces (LIPs). The emplacement of LIPs involved $> 1 \times 10^6$ km³ of mainly basaltic magmas, via eruptions or shallow intrusions, often within sedimentary basins rich in organic and inorganic carbon[4]. LIPs contributed to global climate forcing through both volcanic emissions (e.g., direct release of mainly inorganic carbon from erupted magmas)[5] and thermogenic degassing (e.g., release of organic and inorganic carbon from the intruded sedimentary rocks, due to contact metamorphism)[6]. Significant input of ¹³C-depleted organic carbon into the surface system during mass extinction events is inferred by the sharp negative carbon-isotope shifts preserved in the sedimentary record[7]. The Central Atlantic Magmatic Province (CAMP)[8] is one of Earth's most voluminous LIPs, with $> 3 \times 10^6$ km³ of basaltic magmas emplaced synchronously with the end-Triassic mass extinction (ca. 201.6–201.3 Ma)[9–11]. Direct geological evidence for abundant volcanic $CO_2$ emissions was recently identified within CAMP basaltic lava flows[12]. Conversely, direct geological evidence for degassing from intruded and heated organic-rich sedimentary rocks, hypothesized and modelled for the CAMP[13,14], is still lacking.

Here, we investigate and quantify $CH_4$ preserved within micrometre-sized fluid inclusions (FIs), hosted in shallow basaltic intrusions from the CAMP in Brazilian Amazonia. These findings indicate that CAMP magmatism generated or remobilized large amounts of $CH_4$, which likely contributed to the end-Triassic climatic and biotic crisis.

## Results

The investigated magmatic rocks were sampled in northern Brazil, where about $1 \times 10^6$ km³ of CAMP basaltic magmas, stacked at shallow depths within the Amazonas and Solimões Basins[13,15,16], formed some of Earth's largest magmatic sills. The analysed samples are from several sills emplaced at about 1–4 km depth (on average corresponding to ca. 50 MPa) in the intracratonic Amazonas Basin[17,18], and outcropping over $> 500$ km distances in the State of Pará, northern Brazil (Fig. 1; Supplementary Fig. 1; Supplementary Data 1). These CAMP sills display U–Pb zircon ages ranging from $201.525 \pm 0.065$ to $201.348 \pm 0.034$ Ma[11,19], overlapping in time with the end-Triassic interval of climatic and environmental changes, carbon cycle disruption and biotic crisis[10,20]. According to geophysical data and borehole investigations, CAMP basaltic magmas intruded the Amazonas and Solimões Basins at different levels, forming 3–7 sills with a cumulative thickness of up to 1 km[18,21]. The CAMP sills intruded Devonian to Carboniferous sedimentary formations of continental and marine origin, including voluminous organic-rich shales and evaporites[22,23]. During CAMP emplacement both the Amazonas and Solimões Basins had a high potential for hydrocarbons, which were either newly generated or remobilized by the thermal anomaly associated with the intrusion of sills[13,17,22]. The primary hydrocarbon generation was related to the burial of organic matter within the subsiding basin from the Late Carboniferous to the Early Triassic, with a peak during the Late Permian, whereas the secondary hydrocarbon generation was related to the thermal anomaly induced by the CAMP magmatic intrusions at the end-Triassic[17,22,24]. Evidence for secondary thermal cracking of source rock organic matter after normal sediment diagenesis and for unconventional $CH_4$ generation through the oxidative dissolution of siderite in

the Solimões Basin[25] highlights the key role of CAMP activity in these hydrocarbon systems.

Our in situ analytical data (see Methods) show that multiphase FIs, entrapped in minerals of the sill-forming magmatic rocks, represent direct evidence of volatiles mobilized and degassed during magma emplacement (Figs. 2, 3; Supplementary Figs. 2, 3).

**Magmatic rocks**. The sill-forming magmatic rocks, geochemically classified as basalts and basaltic andesites, consist of microgabbros (dolerites), mainly composed of plagioclase, clinopyroxene, Fe-Ti oxides, olivine (when quartz is absent) and interstitial micrographic intergrowths of quartz and alkali feldspar, along with amphibole, biotite and apatite[15,23,26–28]. Three different crystallization stages are identified on the basis of microstructural position and reciprocal equilibria of mineral phases (Fig. 2; Supplementary Note 1). Early-magmatic minerals (70–90 vol.%), such as plagioclase, clinopyroxene and oxides, plus rare olivine in quartz-free samples, constitute the main crystal framework of the sills (Fig. 2a). Late-magmatic minerals (10–30 vol.%), mainly hydrous silicates, feldspars and quartz, often occupy interstitial domains (Fig. 2b). Hydrothermal minerals, like carbonate, quartz and sulfides among others, generally replace the magmatic minerals or crystallize in veins crosscutting the microgabbros (Supplementary Fig. 4). Quartz is the main host mineral phase for FIs and crystallizes during both late-magmatic and hydrothermal stages. Late-magmatic quartz occurs interstitially in single crystals or aggregates, with eu- to an-hedral crystal habit, or in irregular intergrowths with alkali feldspar as graphic textures (Fig. 2c). It represents on average 5 vol.% (up to 13 vol.%) of the investigated microgabbros, as assessed by microscope observations and consistent with calculated normative compositions (Supplementary Data 2).

**Fluid inclusions**. Primary FIs (i.e., those entrapped during crystallization of the host mineral) are abundant within late-magmatic and hydrothermal quartz (Fig. 3; Supplementary Note 2). These FIs are either vapour- or liquid-rich, measure 1–50 μm in their maximum dimension and are randomly distributed within all investigated crystals (Fig. 3a). In the coexisting vapour- and liquid-rich FIs hosted by late-magmatic quartz, the liquid phase is $H_2O$ and the vapour phase is $CH_4$, which has a measured density of about 0.02 g/cm³ (Figs. 3b, c, 4; Supplementary Data 3; Source Data 1, 2). Halite (NaCl) is the only identified solid phase occurring as cubic crystals in some liquid-rich FIs, which have a salinity of $> 26$ wt.%, calculated after ref. [29]. In the mainly liquid-rich FIs hosted by hydrothermal quartz, both liquid and vapour phases are $H_2O$.

**Geothermobarometry**. Quartz crystallization and FI entrapment occurred during the late-magmatic to hydrothermal stages (Supplementary Note 3). The crystallization temperatures of quartz, calculated from its titanium content[30], are mainly in the range 700–600 °C for the late-magmatic quartz, with generally higher values at crystal cores, and mainly in the range 500–400 °C for the hydrothermal quartz (Fig. 5; Supplementary Fig. 5; Supplementary Data 4, 5).

**Magma–sediment interaction**. Primitive CAMP basalts of both low-Ti ($TiO_2 < 2.0$ wt.%) and high-Ti ($TiO_2 > 2.0$ wt.%) types contain olivine[31,32] and are thus not compatible with the presence of quartz. Hence, significant enrichment in $SiO_2$ is required to achieve quartz saturation and crystallization in the Amazonian CAMP sills[15] (Fig. 6a, b). Starting from typical low-Ti[33] and high-Ti[31] CAMP basalts with 0.5 wt.% $H_2O$[12] (Supplementary Data 2),

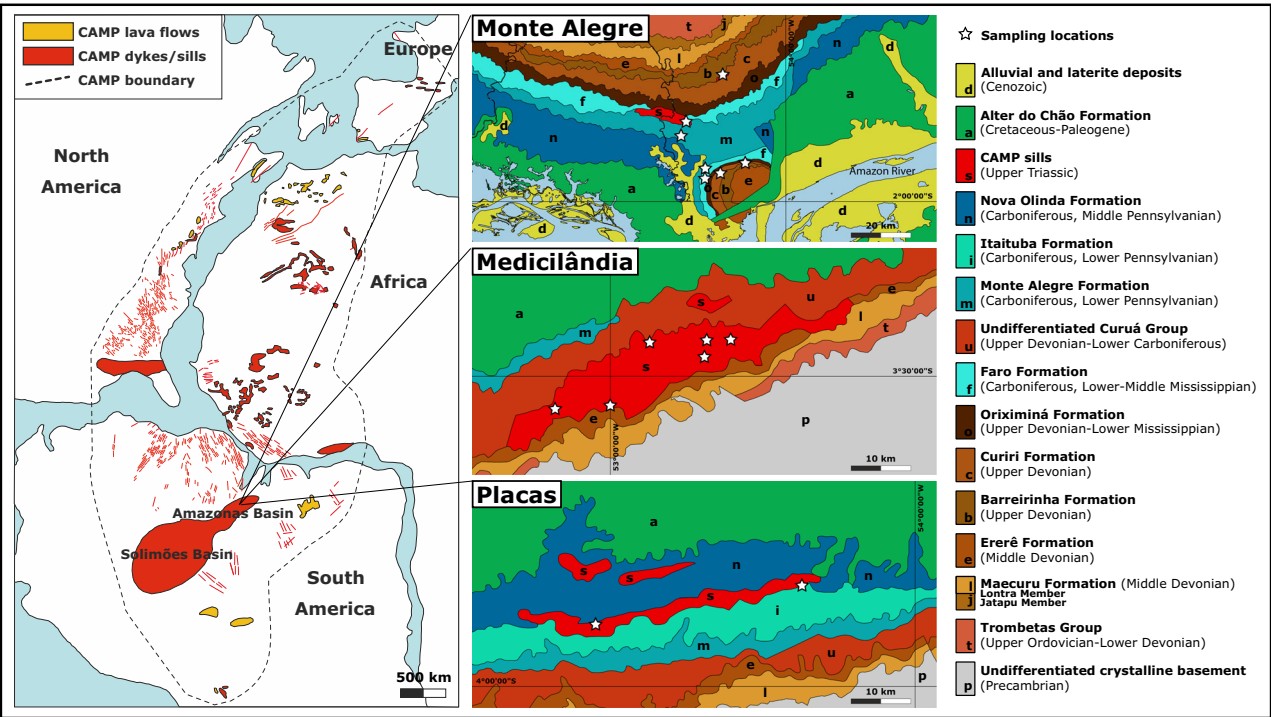

**Fig. 1 Sketch map of the CAMP with sampling locations in the Amazonas Basin.** The geological map of CAMP at about 201 Ma is modified after ref. [16] and references therein, and the geological maps of Monte Alegre, Medicilândia and Placas (Amazonas Basin) are modified after ref. [23]. The CAMP sills, intruding Devonian to Carboniferous sedimentary formations and locally named Penatecaua Diabase, outcrop as km-scale elongated intrusive bodies, approximately displaying an E–W orientation[23]. Among the sedimentary formations and groups intruded by sampled sills, the Urupadi Group includes the Maecuru Formation (Lontra Member; delta fan siliciclastic rocks) and the Ererê Formation (tidal plain siliciclastic rocks). The Curuá Group includes the Barreirinha Formation (mainly black shales deposited in an anoxic marine setting), the Curiri Formation (glacial siliciclastic rocks), the Oriximiná Formation (fluvio–deltaic siliciclastic rocks, with a glacial contribution) and the Faro Formation (fluvio–deltaic siliciclastic rocks), and is locally represented by the undifferentiated Curuá Group (black shales and other siliciclastic rocks of marine, glacial and fluvio–deltaic origin). The Tapajós Group includes the Monte Alegre Formation (siliciclastic and carbonate rocks from fluvio–aeolian and marine environments), the Itaituba Formation (siliciclastic rocks with carbonate intercalations from shallow marine and intertidal plain environments) and the Nova Olinda Formation (siliciclastic rocks at the base, evaporites and limestone lenses at the top, from fluvio–lacustrine and hypersaline restricted marine environments). During CAMP activity hydrocarbons were newly generated or remobilized from pre-existing pools in both Amazonas and Solimões Basins[22]. The source rocks are mainly represented by marine black shales constituting the lower part of the Barreirinha Formation[17,22,24]. The reservoir rocks are mainly represented by neritic–deltaic sandstones, siltstones and shales of the Maecuru and Ererê Formations, by glacio–marine diamictites, shales and siltstones of the Curiri Formation, by aeolian sandstones, lacustrine and interdune shales of the Monte Alegre Formation and by lenticular sandstones of the Nova Olinda Formation[17,22].

Rhyolite-MELTS modelling[34] (see Methods) suggests that the observed, relatively abundant quartz crystallization (ca. 5 vol.%) requires at least 10 wt.% assimilation of silica-rich rocks such as shales (Supplementary Data 6). Shales, which are present in the intruded Paleozoic sedimentary sequence (Fig. 1; Supplementary Fig. 1), would dehydrate from ca. 500 °C and melt from ca. 700 °C[35,36], and their assimilation would also enhance zircon saturation, consistently with the relative abundance of zircon among the late-magmatic minerals of quartz-bearing basaltic rocks in the Amazonian CAMP sills[11,19]. Crustal assimilation is particularly relevant for the high-Ti magmas (> 50 % of the investigated samples; Supplementary Data 2), which would not crystallize any magmatic quartz without contamination (Supplementary Data 6). According to our petrologic modelling, the contaminated basaltic magma contains relatively large amounts of $H_2O$ (ca. 0.9 wt.%), suggesting that the entire volume of Amazonian CAMP sills released about $2.7 \times 10^4$ Gt $H_2O$.

**Methane and halite in aqueous fluids.** In addition to $H_2O$, the amounts of $CH_4$ and NaCl in the Amazonian CAMP sills during late-magmatic quartz crystallization are estimated using our microscopy and microspectroscopy data (see Methods). These data combined with experimental phase equilibria in the ternary $H_2O$–NaCl–$CH_4$ system[37], at a pressure and temperature consistent with those of late-magmatic quartz crystallization, suggest that in the Amazonian CAMP sills the $H_2O/CH_4$ molar ratio was about 3.3 and the $H_2O/NaCl$ molar ratio was about 2.6 (Fig. 6c). Therefore, the total amount of $CH_4$ ($7.2 \times 10^3$ Gt; $4.5 \times 10^{17}$ mol) and NaCl ($3.4 \times 10^4$ Gt; $5.8 \times 10^{17}$ mol) for all Amazonian CAMP sills during late-magmatic quartz crystallization hinges on the estimated amount of $H_2O$. Potentially, similar or even larger amounts of fluid phases were also mobilized from the intruded sedimentary sequence before the crystallization of late-magmatic quartz, but were not preserved by FIs in other mineral phases within the sills.

**Discussion**

The rapid and pulsed intrusion of about $1 \times 10^6$ km³ CAMP basaltic magmas in Amazonia likely produced widespread and prolonged heating within the shallow sedimentary basins[11,19]. The magmatic interaction with sedimentary rocks such as shales and evaporites may account for the excess chlorine observed in the analysed FIs and biotites from Amazonian CAMP sills[28]. The formation of thermogenic $CH_4$ requires relatively low temperatures, from ca. 60 to 300 °C[38], and may occur within the meta-morphic aureoles, which extend up to 250% of sills thickness[39].

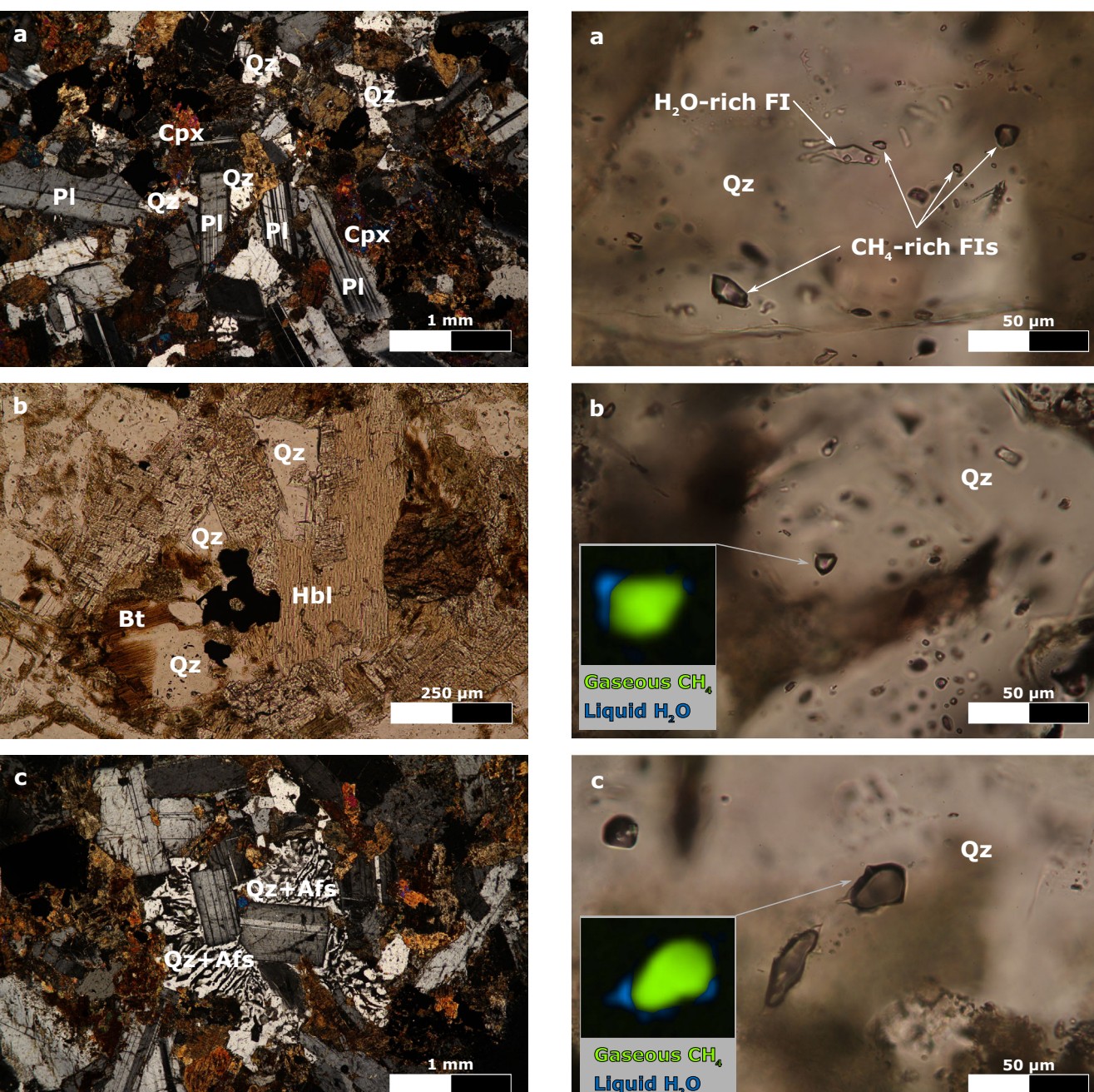

**Fig. 2 Photomicrographs of microgabbros. a** Microgabbro with early-magmatic plagioclase (Pl) and clinopyroxene (Cpx), and interstitial late-magmatic quartz (Qz; sample RP125). **b** Late-magmatic quartz, brown hornblende (Hbl) and red biotite (Bt; sample RP125). **c** Graphic texture between late-magmatic quartz and alkali feldspar (Qz+Afs; sample RP130). Transmitted, crossed polarized light (**a**, **c**) and transmitted, plane polarized light (**b**).

**Fig. 3 Photomicrographs of FIs, along with hyperspectral Raman maps of CH$_4$-rich FIs. a** H$_2$O-rich FI, containing 3 phases (i.e., liquid + vapour + solid phases), and CH$_4$-rich FIs, containing 2 phases (i.e., liquid + vapour phases), hosted in late-magmatic quartz (Qz; sample RP128). **b**, **c** CH$_4$-rich FI, hosted in late-magmatic quartz (samples RP125 and RP128), along with the corresponding hyperspectral Raman map, where the vapour phase is CH$_4$ (green phase) and the liquid phase is H$_2$O with ca. 10 wt.% salinity (blue phase). Transmitted, plane polarized light.

Massive amounts of fluids in the H$_2$O–NaCl–CH$_4$ system were likely released during heating of the intruded Paleozoic sedimentary sequence up to its melting temperature (ca. 700 °C; Fig. 6a). Favoured by temperature and pressure gradients, H$_2$O–NaCl–CH$_4$ fluids migrated after being generated through devolatilization reactions and organic cracking within the organic-rich sedimentary basins[28,39,40]. Furthermore, CH$_4$ already present within Paleozoic reservoir rocks at 201 Ma was remobilized due to CAMP intrusions, which likely induced additional maturation of organic matter, forming new CH$_4$ within the Amazonian sedimentary basins[17,22].

A dominant deep magmatic origin of CH$_4$ is not likely since its maximum solubility in basaltic magmas at low oxygen fugacity is very low (< 0.1 wt.%)[41] and melt inclusions from CAMP basaltic rocks indicate exsolution of magmatic carbon mainly as CO$_2$ at greater depths, before the magmas reached the shallow sedimentary basins[12]. Similarly, the lack of graphite and native iron in

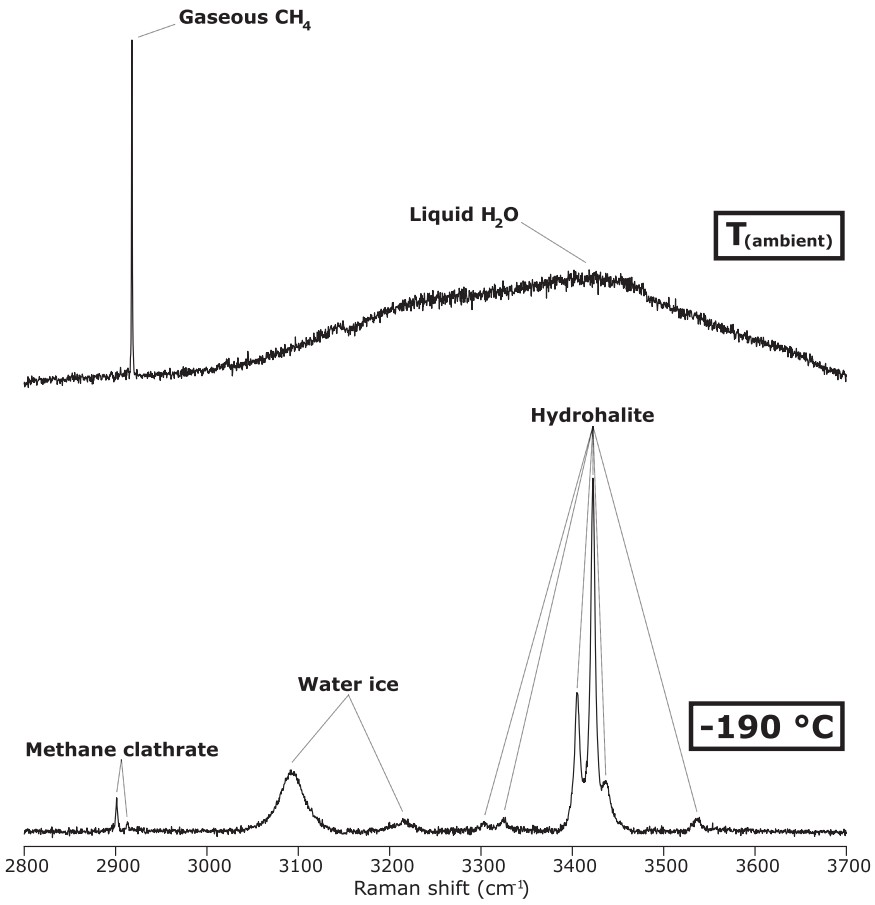

**Fig. 4 Raman spectra of the investigated phases within late-magmatic quartz-hosted FIs.** Raman spectrum of gaseous $CH_4$ and liquid $H_2O$ within a liquid-rich FI (sample RP125), acquired at ambient temperature by confocal Raman microspectroscopy (uppermost Raman spectrum). Gaseous $CH_4$ corresponds to the intense and sharp Raman band at ca. 2917 $cm^{-1}$, and liquid $H_2O$ corresponds to the broad Raman band ranging from 3000 to 3700 $cm^{-1}$. Raman spectrum of methane clathrate, water ice and hydrohalite (solid phases formed in the $H_2O$–NaCl–$CH_4$ system under freezing conditions) within a liquid-rich FI (sample RP116), acquired at −190 °C by confocal Raman microspectroscopy combined with microthermometry (lowermost Raman spectrum). Methane clathrate corresponds to the sharp Raman band at ca. 2904 $cm^{-1}$ and to the weak Raman band at ca. 2916 $cm^{-1}$, water ice corresponds to the intense Raman band at ca. 3100 $cm^{-1}$ and to the weak Raman band at ca. 3223 $cm^{-1}$, and hydrohalite corresponds to several Raman bands at ca. 3303, 3325, 3405, 3423, 3436 and 3540 $cm^{-1}$.

the analysed microgabbros and the presence of $H_2O$–NaCl–$CH_4$ fluids in equilibrium with these rocks rule out direct magmatic assimilation of organic matter to form $CH_4$[42].

From source or reservoir rocks, $CH_4$-bearing fluids migrated into the crystallizing gabbroic sills, with records of their presence entrapped by FIs, before potentially reaching the atmosphere[13]. Decompression and cracking induced by contraction of the cooling magma bodies may have promoted the migration of fluids towards the sills[28], favouring their partial entrapment within the crystallizing quartz during the late-magmatic stage, consistent with $CH_4$ stability up to 1200 °C in both C–O–H and C–O–H–S systems[42,43].

The large amount of $CH_4$ ($4.5 \times 10^{17}$ mol) estimated from the FI analysis is consistent with the emissions of carbon ($2.0 \times 10^{18}$ mol) modelled for the contact metamorphism induced by CAMP intrusions within the Amazonian basins[13]. The release of carbon species due to thermogenic degassing from carbon-rich sedimentary sequences is also suggested by the presence of similar $CH_4$-bearing FIs and hydrothermal vents from the North Atlantic LIP[6,44], likely responsible for the Palaeocene–Eocene Thermal Maximum[45], and was thermodynamically modelled for several LIPs[46,47]. A modern analogue of the Amazonian CAMP magma–sediment interaction, even if on a much smaller scale, is represented by the Lusi mud-eruption (NE Java, Indonesia). Lusi

is a sediment-hosted hydrothermal system, where shallow magmatic intrusions within a sedimentary sequence affect source rocks and/or hydrocarbon reservoirs, degassing large amounts of both volcanic and thermogenic carbon species into the atmosphere[48].

The direct evidence for magma–sediment interaction recorded by FIs of the investigated CAMP sills indicates that large amounts of $CH_4$ were generated or remobilized due to the end-Triassic magmatism. A minimal part of this volatile load was entrapped within quartz-hosted FIs. The majority was either reintroduced into the sedimentary host rocks, forming exploitable $CH_4$ reservoirs in structural and lithological traps[17], or discharged into the surface system during CAMP activity[11], possibly through hydrothermal vents and pipes like those detected in other LIPs[6,49]. Methane is among the highest impact greenhouse gases[50] and chlorine, which is mostly dissolved in $H_2O$ associated with $CH_4$, can induce ozone depletion[51]. In addition to the $CO_2$ exsolved from CAMP magmas in the deep transcrustal magmatic system[12], the $CH_4$ released from the shallow sedimentary basins likely contributed to the end-Triassic global warming of up to 4 °C[52], forcing the global climate changes and triggering the mass extinction. The injection of sedimentary organic $CH_4$, displaying a $^{13}C$-depleted isotopic signature, into the exogenic carbon cycle may have also produced the large negative carbon-isotope shifts

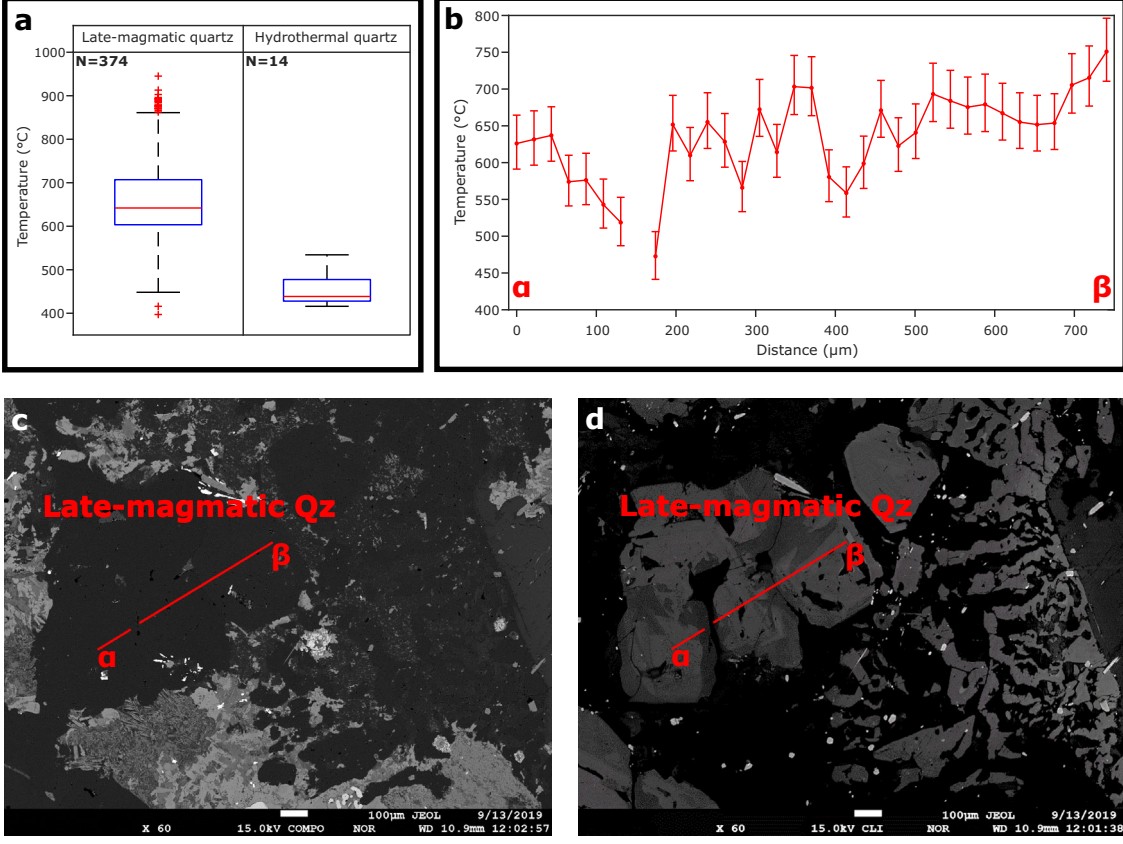

**Fig. 5 Crystallization temperatures of quartz. a** Calculated temperature ranges of both late-magmatic and hydrothermal quartz (for all the analysed samples) in box-and-whisker plots. In box-and-whisker plots, the middle line of the box indicates the median, the bottom and top edges of the box show the lower and upper quartiles respectively, the whiskers extend to the minimum and maximum data that are not outliers, and the outliers are plotted individually. For each temperature datum, the average total uncertainty is ±40 °C for late-magmatic quartz and ±32 °C for hydrothermal quartz. Note that the data displaying below-detection-limit values or > 15 % analytical errors for Ti or displaying >5000 ppm values for Al are not plotted. **b** Temperature transect (α–β; panels **c** and **d**) of late-magmatic quartz crystals (sample RP136), based on TitaniQ thermobarometry[30]. For each calculated temperature, the error bar represents the total uncertainty, which is given by the sum of internal (depending on analytical measurements) and external (depending on both crystallization pressure and $TiO_2$ activity relative to rutile saturation) uncertainties. **c**, **d** Backscattered-electron (BSE) and greyscale cathodoluminescence (CL) images of the analysed late-magmatic quartz (Qz) crystals, displaying normal growth zonation, with high-Ti (CL-brighter) cores and low-Ti (CL-darker) rims.

in the sedimentary record[14]. Our observations highlight the role of LIP magmatism and its interactions with rocks and fluids within sedimentary basins in controlling global climate changes and the evolution of life on Earth, as well as in enhancing hydrocarbon maturation and remobilization.

## Methods
**Sample selection and preparation**. The microgabbros analysed for this study are from CAMP sills intruded in the Amazonas Basin (Fig. 1). Mainly based on the variability in mineralogical composition and hydrothermal alteration, 20 out of a suite of 50 intrusive rock samples were selected from the outcrops near the villages of Monte Alegre, Medicilândia and Placas in the State of Pará, northern Brazil (Fig. 1; Supplementary Fig. 1; Supplementary Data 1). These outcrops mainly occur along road cuts and in quarries, and are generally scattered and not continuous, due to the flat topography and dense vegetation. Within the Paleozoic sequence of the Amazonas Basin, the sills sampled at Monte Alegre, north of the Amazonas River, intruded both the Curuá Group (Devonian–Carboniferous) and the Tapajós Group (Carboniferous)[23]. The sill sampled at Medicilândia, south of the Amazonas River, intruded between the Urupadi Group (Devonian) and the Curuá Group (Devonian–Carboniferous)[23]. Lastly, the sill sampled at Placas, south of the Amazonas River, intruded the Tapajós Group (Carboniferous)[23]. From the screened microgabbros, the 10 most representative FI-bearing samples were characterized in detail by optical microscopy and confocal Raman microspectroscopy (Supplementary Data 1). Glue-free double-polished thick (about 100 μm) sections were employed for in situ confocal Raman microspectroscopy, in order to avoid contamination and signal interferences from any artificial carbon-bearing compounds.

**X-ray fluorescence (XRF) spectrometry**. This analytical technique was employed to determine the whole-rock chemical composition (major elements) of the most representative FI-bearing samples (Supplementary Data 1). The analyses were conducted at the Department of Geosciences, University of Padova, using a WDS Philips PW2400 sequential spectrometer. The analytical uncertainty is < 3% for all major elements.

**Microthermometry**. This analytical technique was employed to investigate the unexposed liquid-rich FIs through the phase transition temperatures, mainly of last-melting and homogenization, and to determine additional phases, in combination with confocal Raman microspectroscopy, under freezing conditions (e.g., hydrohalite; Supplementary Data 1). The analyses were conducted at the Lithosphere Fluid Research Lab and at the Research and Industrial Relations Center of the Faculty of Science, Eötvös Loránd University of Budapest, using a LINKAM THMS600 heating–freezing stage, mounted on a NIKON Eclipse E600 petrographic microscope. The investigated FIs were cooled down to −190 °C using liquid $N_2$, and then heated until homogenization was reached, potentially up to 600 °C. A NIKON × 50 objective was used to observe the phase transitions.

**Confocal Raman microspectroscopy**. This analytical technique was employed to characterize the fluid and solid phases of unexposed FIs (Supplementary Data 1). The analyses were conducted at the Research and Industrial Relations Center of the Faculty of Science, Eötvös Loránd University of Budapest, using a HORIBA Jobin Yvon LabRAM HR 800 Raman microspectrometer. Both spot and areal analyses were carried out below the sample surface for all the phases within unexposed FIs. Spot Raman analysis allowed to acquire the spectra of fluid and solid phases, and to characterize them through their spectral features. Areal Raman analysis (i.e., Raman mapping) allowed to reconstruct and image the spatial distribution of the

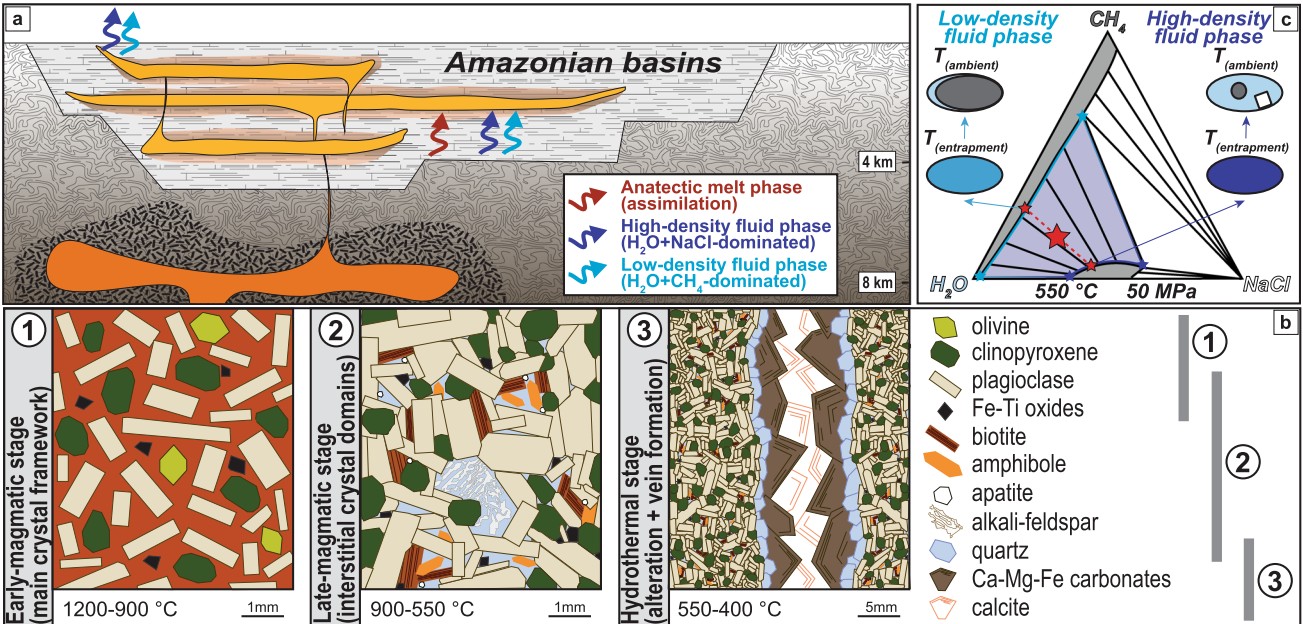

**Fig. 6 Sketch of CAMP sills emplacement, crystallization and fluids entrapment. a, b** Magmatic crystallization started in rising magmas within the deep transcrustal magmatic system, below the basins, and continued within the sills. One anatectic melt phase and two immiscible fluid phases from the heated organic-rich sedimentary rocks migrated within the sills during the late-magmatic crystallization. Hydrothermal alteration and vein formation occurred after the magmatic crystallization. **c** Coexisting fluid phases, occasionally entrapped as FIs mainly in crystallizing quartz, are shown at entrapment conditions of 550 °C and 50 MPa (modified after ref. [37]). Two immiscible fluid phases, dominated by $H_2O+NaCl$ and $H_2O+CH_4$ respectively (small red stars), coexist along tie-lines for the average composition of a bulk fluid system (large red star) in the highlighted 2-phase field. When the FIs cooled to ambient temperature, $CH_4$ bubbles nucleated and NaCl crystals precipitated.

same phases within unexposed FIs. A frequency doubled Nd–YAG green laser with a 532 nm excitation wavelength was employed, displaying 120 mW at the source and 23 mW on the sample surface, and a He–Ne red laser with a 633 nm excitation wavelength was employed exclusively to analyse $H_2O$ within calcite-hosted FIs. An OLYMPUS × 100 (N.A. = 0.9) objective was used to focus the laser on the analysed sites. Raman spectra acquisition was conducted in both single- and multi-window settings, from ambient temperature down to −190 °C, combined with a LINKAM THMS600 microthermometric stage. For spot Raman analysis, a 100 μm confocal hole, a 1800 grooves/mm optical grating, 2–3 accumulations and a 30–60 s exposition time were employed. For areal Raman analysis, a 100 μm confocal hole, a 600 grooves/mm optical grating, 1–20 accumulations and a 1–30 s exposition time were employed. The investigated spectra range from 70 to 4000 cm$^{-1}$, depending on the spectral region of interest for each analysed phase. The spectral resolution of measurements varies from 0.8 to 3.0 cm$^{-1}$ for the spot spectra, and from 2.4 to 3.0 cm$^{-1}$ for the maps. Spot Raman data were processed through LabSpec 5 software, and areal Raman data were processed through both LabSpec 5 and LabSpec 6 softwares.

**Electron microprobe (EMP) analysis.** This analytical technique was employed to measure the Ti and Al concentrations in both late-magmatic and hydrothermal quartz (Supplementary Data 1). The analyses were conducted at the Department of Mineral Sciences, Smithsonian Institution, Washington DC, using a JEOL 8530F hyperprobe. The beam current was set at 200 nA, the accelerating voltage at 15 kV, and the spot diameter at 10 μm. The Ti concentrations were measured using 3 PETL spectrometers, and the Al concentrations using 2 TAP spectrometers, for 300 s on the peak and 15 s on the background, yielding a detection limit of 6–8 ppm for Ti and 4–6 ppm for Al. As standards, ilmenite was used for Ti and corundum for Al.

**Cathodoluminescence (CL) analysis.** This analytical technique was employed to distinguish late-magmatic and hydrothermal quartz (Supplementary Data 1). Colour CL images were acquired at the C.N.R., Institute of Geosciences and Georesources, Padova, using a Cold Cathode Luminescence 8200 mk3. The beam current was set at 220 μA and the accelerating voltage at 20 kV. Greyscale CL images were acquired at the Department of Mineral Sciences, Smithsonian Institution, Washington DC, using a JEOL panchromatic cathodoluminescence detector. The beam current was set at 20 nA and the accelerating voltage at 15 kV.

**Petrologic modelling of magma crystallization.** The equilibrium crystallization of CAMP sills was modelled in open and closed system, using Rhyolite-MELTS (code version 1.0.2)[34]. We started from typical low-Ti (sample AN133)[33] and

high-Ti (sample M13)[31] CAMP basalts with 0.5 wt.% $H_2O$[12] in the initial magma, and considered an average shale with 5.0 wt.% $H_2O$[53] as the assimilated crustal material (Supplementary Data 2, 6). The initial temperature of the assimilated shale was set at 100 °C, and its amount was set at 10 wt.% for low-Ti magmas and at 20 wt.% for high-Ti magmas. The oxygen fugacity was set at the FMQ buffer, and the pressure generally at 50 MPa, corresponding to the average intrusion depth of the investigated CAMP sills. Our petrologic modelling suggests that at least 10 wt.% crustal assimilation is necessary to reproduce the observed amount of quartz within the Amazonian CAMP sills. Furthermore, high-Ti CAMP basalts do not reach quartz saturation without shale assimilation and up to 20 wt.% crustal assimilation is necessary to reproduce the same observed amount of quartz.

**Quantification of $H_2O$, NaCl and $CH_4$.** During the late-magmatic stage, at ca. 900–550 °C and 50 MPa, the system was likely formed by at least one exsolved fluid phase ($H_2O$–NaCl–$CH_4$) coexisting with a silicate melt, from which quartz, alkali feldspar and hydrous mineral phases, such as amphibole and biotite, crystallized. Within late-magmatic quartz, liquid- and vapour-rich FIs are associated together without any distinctive distribution between crystal cores, mantles and rims, and thus constitute a peculiar FI assemblage, that is typical of shallow, silicic, magmatic/hydrothermal environments[54]. This FI assemblage reveals that two different, immiscible fluid phases were present during the late-magmatic stage, and were occasionally trapped into crystallizing quartz as FIs. In respect of Roedder's rules[54], the liquid-rich FIs likely trapped a single homogeneous $H_2O$+NaCl-dominated fluid phase, containing minor amounts of $CH_4$ (i.e., high-density fluid phase), whereas the vapour-rich FIs likely trapped a single homogeneous $H_2O$+$CH_4$-dominated fluid phase, containing minor amounts of NaCl (i.e., low-density fluid phase). Considering the complex $H_2O$–NaCl–$CH_4$ ternary system (Fig. 6c), broad regions of immiscibility are present at pressures and temperatures consistent with the entrapment conditions of the investigated FIs[37,55–57]. For any bulk composition that lies in the largest 2-phase field at these conditions, one high-density fluid phase and one low-density fluid phase coexist[55,56]. The high-density fluid phase is enriched in $H_2O$ and NaCl (i.e., $H_2O$+NaCl-dominated fluid) and corresponds to our 2 or 3 phase-bearing, liquid-rich FIs at ambient temperature. The low-density fluid phase is enriched in $H_2O$ and $CH_4$ (i.e., $H_2O$+$CH_4$-dominated fluid) and corresponds to our vapour-rich FIs at ambient temperature. The presence of NaCl and the low pressure of crystallization promote fluid immiscibility, and the 1-phase field, depending on both pressure and temperature, is generally very restricted[37].

The FIs hosted in late-magmatic quartz may contain $H_2O$+NaCl+$CH_4$, $H_2O$+$CH_4$ or pure $CH_4$. Therefore, $CH_4$ is the only component always present during the late-magmatic stage. However, this variability in the chemical composition of the investigated FIs reveals a continuous compositional evolution of the fluid phases throughout the whole temperature range for the crystallization of

late-magmatic quartz. For instance, the addition of further exsolved fluids from rising magmatic recharges, as well as of brines from intruded sedimentary rocks, and the crystallization of hydrous mineral phases along with quartz may progressively modify the composition of the complex $H_2O–NaCl–CH_4$ system.

The total volume of ca. $1.0 \times 10^6$ $km^3$ for CAMP sills in Amazonia[16] corresponds to a total mass of ca. $3.0 \times 10^6$ Gt, assuming a density of 3.0 $g/cm^3$ for gabbroic rocks[58]. Our petrologic modelling suggests that at least $2.7 \times 10^5$ Gt of shales may have been assimilated during the emplacement of the Amazonian CAMP sills. According to our Rhyolite-MELTS modelling, the $H_2O$ dissolved in the melt at quartz saturation is about 2.2 wt.%, consistently with experimental data on granitic systems[59,60], and the $H_2O$ exsolved from the crystallizing sills is about 0.9 wt.%. Therefore, the total mass of $H_2O$ released from the Amazonian CAMP sills is about $2.7 \times 10^4$ Gt, equivalent to $1.5 \times 10^{18}$ mol. However, the higher amount of hydrous mineral phases and the lower crystallization temperatures of quartz in the investigated microgabbros compared to our modelling results, suggest that the late-magmatic stage was likely richer in $H_2O$, thus not derived exclusively from the magmatic (i.e., juvenile and assimilated) component considered in our petrologic modelling, but possibly also from sedimentary and metamorphic components originated in the host sedimentary sequence. Similarly, the occurrence of pure $CH_4$-bearing FIs hosted in late-magmatic quartz likely represents $CH_4$ fluxes from the metamorphic aureoles into the crystallizing sills.

Experimental phase equilibria in the ternary $H_2O–NaCl–CH_4$ system at 550 °C and 50 MPa[37] indicate that the investigated FI assemblage represents a fluid system in a 2-phase field for pressure and temperature consistent with the entrapment conditions (Fig. 6c). In order to quantify the amount of $CH_4$, we observed that within the vapour-rich FIs at ambient temperature the gaseous $CH_4$ occupies most commonly about 95 vol.% of the whole FI and coexists with liquid $H_2O$ displaying ca. 10 wt.% salinity. Given an average density of ca. 0.02 $g/cm^3$ for gaseous $CH_4$ and of ca. 1.07 $g/cm^3$ for liquid $H_2O$ with 10 wt.% salinity[61], the $H_2O/CH_4$ molar ratio is generally about 2.2 in the vapour-rich FIs. The bulk fluid system is considered to have the average composition between those of the coexisting low- and high-density fluid phases, forming the vapour- and liquid-rich FIs respectively (Fig. 6c). Therefore, the $H_2O/CH_4$ molar ratio was about 3.3 in the bulk fluid system, and the total amount of carbon, in the form of $CH_4$, within the sills during late-magmatic quartz crystallization was approximately $4.5 \times 10^{17}$ mol ($7.2 \times 10^3$ Gt $CH_4$). Similarly, the $H_2O/NaCl$ molar ratio was about 2.6 in the bulk fluid system, and the total amount of chlorine, in the form of NaCl, within the sills during late-magmatic quartz crystallization was approximately $5.8 \times 10^{17}$ mol ($3.4 \times 10^4$ Gt NaCl).

## Data availability

All data generated in this study are provided in the Supplementary Information (Supplementary Notes 1–3, Supplementary Figs. 1–5 and Supplementary References), in the Supplementary Data 1–6 and in the Source Data 1, 2. Source data are provided with this paper.

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

## Acknowledgements

This study was supported by the following collaborative research projects: PRIN 20178LPCP (Italy) to A.M. and A.D.M., TKP2020-IKA-05 (Hungarian Ministry of Human Capacities) to L.E.A. and Cs.Sz., BART_SID19_01 (Italy) to O.B., Young Research Talents Project 301096 MAPLES (Research Council of Norway) to S.C., and NERC Large Grant NE/N018559/1 (UK) to R.J.N. The authors thank A. Sansone (University of Padova) for sample selection, L. Tauro (University of Padova) for sample preparation, and G. Bellieni for valuable assistance during sampling.

## Author contributions

M.C. and A.M. devised the project. A.M., M.E., E.M.G.V. and A.D.M. carried out the field work. M.C., A.M., L.E.A. and M.R.A. collected the analytical data. M.C., A.M., L.E.A., M.R.A., O.B., S.C., J.D.C., M.E., E.M.G.V., A.D.M., R.J.N. and Cs.Sz. contributed to data interpretation and writing of the manuscript.

## Competing interests

The authors declare no competing interests.
