## [Peer Review File · Nature Communications]

Reviewers' Comments:

Reviewer #1:

Remarks to the Author:

Review of "Basalt-methane interaction at the roots of the end-Triassic crisis" by Capriolo et al. for Nature Communications.

The manuscript by Capriolo et al. presents data from fluid inclusions in quartz crystals that are present as late stage, interstitial crystallisation products in CAMP mafic sills. The authors report on methane in the fluid inclusions, which they argue was sourced from the organic-rich sedimentary rocks that were intersected by CAMP sills in the shallow crust. The implications of these results are profound, as they indicate that methane fluxing, triggered by CAMP intrusions, could have added to the total emitted volatile load of the CAMP. These volatiles would have induced climate warming, which in turn could have helped to trigger the end-Triassic mass extinction.

The hypothesis that volatiles sourced from sedimentary basins could have contributed to the volatile load at LIPs has previously been posited. However, this study is novel in that it brings new insight into our understanding of the processes involved by presenting rare and unusual evidence for crustal methane mobilisation by LIPs. I am not aware of similar evidence in the literature (beyond the previous work looking at CO₂ in gas bubbles in melt inclusions by Capriolo et al., 2020, NCOMMS), so this is really an exciting discovery in my opinion. I am convinced that these results will be of interest to others in the community and the wider field.

I must say that I enjoyed reading this manuscript and found it engaging and the results intriguing. The presence of quartz in these CAMP rocks as well as the methane within them is fascinating and the results bring fresh insight to the field of study of LIPs. The manuscript is well written and the figures are of good quality and useful. There appears to be enough information provided in the supplementary files for this work to be reproducible and the treatment of the data seems appropriate. I have left some comments on the manuscript PDF file with some questions as well as some suggestions for clarification in various places. These comments reflect my thoughts while reading the manuscript and I hope you find them useful.

What's left to say except well done on a very nice piece of work.

Good luck,

Frances Deegan, Uppsala, 8th April 2021.

Reviewer #2:

Remarks to the Author:

Dear authors,

It was a pleasure reading your paper, which is well written, well structured and nicely illustrated. The paper is certainly timely and will surely attract a wide spectrum of readers. It is relevant for all research carried out on LIPs and their impact on palaeoclimate and biota.

I have very few comments and suggestions, which you can find below.

In general, I think your paper could benefit from a few sentences explaining why the methane found in the fluid inclusions is not a just part of the magmatic process. I know that you have stated this in the paper, but not all readers who will be interested in this paper will have insights into magma and contact metamorphic processes. But this is just a suggestion.

The Figures are nice. The supplementary tables are well organized and supplementary information also.

I will recommend minor revision.

Good luck with the review!

Best regards

Sofie Lindström

Specific comments

Line 27. Replace "is" before synchronous with "was".

Line 28–30. Wouldn't it be better to write: "Here we show direct evidence that the interaction between basaltic magmas and the host sedimentary sequence produced methane during the emplacement of this large igneous province".

Line 40. Maybe remove "the" before "Large Igneous Provinces" .or replace it with "two".

Line 45–46. Why not "atmospheric system"?

Line 47–50. I think that here you should mention that thermogenic C release has been hypothesized and modelled by Heimdal et al. 2020. That paper is already cited in your manuscript, but it could be mentioned here.

Line 90–91. Where does the halite come from? I can't find any information about this in the supplement either. Do you expect the halite and the CH₄ to have been generated by the mixture with the same sedimentary rocks?

Line 107–108. It would be good with a reference to back up the last part of the sentence.

Line 108–110.

Line 111–113. This "globally fluxed" seems premature here. Perhaps better to focus on the observations first?

Line 133–136. Has it been proven to have been present already at 201 Ma, or is this a hypothesis?

Line 143. Maybe instead: "CH₄-bearing fluids must have migrated into ... from source or reservoir rocks"

Line 165. I don't like the "surface system" term. Everything above to a certain height the surface is part of the atmosphere.

Line 173–175. I think here you should stress that it is not merely the LIP magmatism but its interactions with rocks, fluids and gases in sedimentary basins, that determines the impact the LIP volcanism will have on the geobiosphere.

Fig. 1. The photos are rather small. Otherwise fine.

Line 321. State why the 20 samples were chosen. Preservation?

Reviewer #3:

Remarks to the Author:

This is a well written study into the petrological evidence for basalt-sediment interactions during the Triassic–Jurassic (TJ) extinction, and the resultant production of methane and other volatiles that are proposed to have subsequently triggered the environmental and biospheric crises at that time. The approach is novel to the best of my knowledge, and the use of fluid inclusions to study these (probably thermogenic) volatile species associated with the Central Atlantic Magmatic Province could potentially be an important step forward in petrological techniques.

However, whilst the technique employed is novel, I'm not sure that the big-picture investigation is impactful enough to warrant publication in Nature Communications. Methane and other thermogenic volatiles linked to CAMP have been proposed as a major contributor to the TJ event for a long time now; the likelihood of the Brazilian basins as a source area has been commented on frequently; and the timing, quantity, and emission rates of these volatiles also modelled. So whilst obtaining petrological support for these volatile emissions is clearly an important iterative step in understanding their contribution to the TJ extinction, I don't think it really advances our overall knowledge of the event that greatly.

Additionally, as a researcher with a limited background in igneous petrology, I found it hard to follow some of the petrological arguments at times. Of course, the length of a Nat. Comms. manuscript is prohibitive in terms of the space for expansive discussion, but given that articles on the major mass extinction events command a wide range of readers with varying interests, I think that it is important to have a little more detail in the main text, so that the arguments can be more easily followed.

In conclusion, whilst I am supportive of the study, and think that it represents a solid piece of

work, I unfortunately do not feel that it is suitable for publication in Nature Communications. I think that it could work well in a slightly longer format journal such as PNAS or EPSL though.

Minor comments:

Abstract: I wouldn't expect there to be citations in the abstract.

L. 24: 'coincide' should be 'coincided'.

L. 26: Change 'changes triggering biotic crises' to 'changes that triggered the biotic crises'.

L. 27: Strictly speaking, the CAMP isn't (or wasn't) an event, it is a geological phenomenon. The voluminous emplacement basalts to form the CAMP was the event.

L. 29: Change 'for' to 'of'. Also, this statement is slightly misleading. The host sedimentary sequences being referred to here are in northern Brazil, which was associated with just one part of the (vast) CAMP. I would imagine that in other areas the country lithology intruded by CAMP basalts were not nearly as volatile rich. So these sedimentary basins should not be stated as being representative of the continental crust intruded by CAMP as a whole.

L. 33: '...unveil the deadly instigator of the global climate change that led to the end-Triassic mass extinction.' There are a couple of things that I don't like with the ending of this sentence. Firstly, I wouldn't say that this study unveils this methane source, as one could argue that it was first unveiled by the well-documented $\delta^{13}\text{C}$ excursions. Secondly, whilst the link between thermogenic volatiles and the extinction is increasingly compelling, I would still suggest that this is a little bit too bold, as we don't yet fully understand to what extent the climate change was driven by thermogenic vs magmatic vs other carbon sources.

L. 38–41: LIP volcanism also coincided with the KPg, and arguably Late Devonian, extinctions. But the link, if any, between volcanism and those two extinctions (and the end-Ordovician) remains unclear, so it's hard to confidently state that all extinctions were caused by LIPs. And lots of LIPs didn't coincide with extinctions. So what point are the authors making here?

L. 42–45: This probably isn't true of all LIPs: at least some of them don't seem to have intruded volatile-rich sedimentary basins (Paraná-Etendeka LIP, for example, and indeed all of the oceanic LIPs).

L. 65: 'basin' should be 'Basin'.

L. 97–98: Something feels wrong to me about giving the temperature ranges with the higher number first. Is this normal in petrology research? If not, I'd switch them to 600–700 and 400–500.

L. 111: What sort of degassing efficiency is assumed in order to generate this volume of emitted H_2O ? 100%? That feels unrealistic though.

L. 129–137: Presumably the generation temperature of thermogenic methane depends on both the depth at which this occurring, and the thermal maturity of the sedimentary rocks prior to intrusion. To what extent is this known? And are there any quantitative constraints on how much methane existed prior to CAMP versus thermogenic production during the LIP emplacement. I know that this is talked about in the supplementary text, but it is quite an important issue that I'd rather see in the main text if possible (another reason that a slightly longer-format manuscript might work better).

L. 144–145: The conventional mechanism by which thermogenic volatiles reach the atmosphere is via breccia pipes formed due to the buildup of pressure by the newly generated volatiles. But here the authors are referring to methane escaping to the atmosphere having first been in the magma. So, are they proposing that magmatic release of thermogenic volatiles was the major mechanism

of their emission instead of breccia pipe explosions? Or a supplementary route? And if the latter, do they have any ideas regarding what percentage of thermogenic volatiles were emitted via magma eruptions versus breccia pipes?

L 154: change 'is' to 'has been'.

L. 156–159: This is an interesting modern analogue, which I wasn't aware of previously. I'm curious as to why the authors also don't mention Vesuvius and/or Etna here? OK, I know that they erupt through carbonates rather than muds, so there isn't a directly analogous production of thermogenic methane, but it's still another well-known modern example of thermogenic volatiles increasing the carbon output of a volcanic system following magma-sediment interaction.

L. 166–167: See earlier point. How much thermogenic carbon was released to the atmosphere via pipe systems vs through magmatic degassing following assimilation of the sedimentary rock into the magmas?

Capriolo et al., “Massive methane fluxing from magma-sediment interaction in the end-Triassic Central Atlantic Magmatic Province”

Response to Reviewers' comments.

All comments are reproduced below and our responses follow in bold text. Corrections are showed as tracked changes in the manuscript, and the comments added on the manuscript PDF file by Reviewer #1 are reported below, in order to answer them as for the comments by the other Reviewers.

Before answering specifically to each comment, we would like to clarify that the original length and layout were set according to *Nature* guidelines, now changed according to *Nature Communications* guidelines, and we would like to thank all Reviewers for their effort in improving our manuscript.

Reviewer #1 (Remarks to the Author):

Review of “Basalt-methane interaction at the roots of the end-Triassic crisis” by Capriolo et al. for Nature Communications.

The manuscript by Capriolo et al. presents data from fluid inclusions in quartz crystals that are present as late stage, interstitial crystallisation products in CAMP mafic sills. The authors report on methane in the fluid inclusions, which they argue was sourced from the organic-rich sedimentary rocks that were intersected by CAMP sills in the shallow crust. The implications of these results are profound, as they indicate that methane fluxing, triggered by CAMP intrusions, could have added to the total emitted volatile load of the CAMP. These volatiles would have induced climate warming, which in turn could have helped to trigger the end-Triassic mass extinction.

The hypothesis that volatiles sourced from sedimentary basins could have contributed to the volatile load at LIPs has previously been posited. However, this study is novel in that it brings new insight into our understanding of the processes involved by presenting rare and unusual evidence for crustal methane mobilisation by LIPs. I am not aware of similar evidence in the literature (beyond the previous work looking at CO₂ in gas bubbles in melt inclusions by Capriolo et al., 2020, NCOMMS), so this is really an exciting discovery in my opinion. I am convinced that these results will be of interest to others in the community and the wider field.

I must say that I enjoyed reading this manuscript and found it engaging and the results intriguing. The presence of quartz in these CAMP rocks as well as the methane within them is fascinating and the results bring fresh insight to the field of study of LIPs. The manuscript is well written and the figures are of good quality and useful. There appears to be enough information provided in the supplementary files for this

work to be reproducible and the treatment of the data seems appropriate. I have left some comments on the manuscript PDF file with some questions as well as some suggestions for clarification in various places. These comments reflect my thoughts while reading the manuscript and I hope you find them useful.

What's left to say except well done on a very nice piece of work.

Good luck,

Frances Deegan, Uppsala, 8th April 2021.

Title: Something feels not quite right with the title. The magma-sediment interaction described in the paper occurred in the "roots" (=plumbing system) of the CAMP. The end-Triassic crisis doesn't really have "roots" in this sense. I'd suggest to change the title to more accurately reflect the paper contents. This will help it to be better discovered and utilized by the community!

We originally assigned the title "Basalt-methane interaction at the roots of the end-Triassic crisis", in order to link the generation and remobilization of methane due to magma-sediment interaction with the onset of end-Triassic global climate change. The term "roots" referred not only to the shallowest part of CAMP plumbing system, where magma-sediment interaction occurred, but also to the potential origin of the end-Triassic mass extinction. In order to disambiguate the previous point, the title has been modified as "Massive methane fluxing from magma-sediment interaction in the end-Triassic Central Atlantic Magmatic Province".

Abstract: Perhaps double-check the journal guidelines to see if the abstract should have references. It seems a little unusual to me.

Apart from that, I must say I was intrigued to read about quartz in your samples! So the abstract has definitely hooked me in.

We thank the Reviewer for this appreciation. The references have been removed from the abstract, according to the journal guidelines.

L. 36-38: This is quite a significant statement and I wonder if it would merit a few more references? I notice that you have the space within the journal limits, so why not add some more?

Two more references added (Wignall, 2015; Ernst & Youbi, 2017).

L. 51: Maybe avoid statements of significance just yet...and bring it up later?

Changed as suggested.

L. 71: I'm not sure that data can be described as "multidisciplinary". Suggest to delete this word.

Deleted as suggested.

L. 72: Are FI's technically entrapped in specific minerals as opposed to microgabbros? I'd suggest to write "entrapped in minerals in the sill-forming microgabbros".

Changed as "entrapped in minerals of the sill-forming magmatic rocks".

L. 76: You don't need to say "quartz-free samples" here – it is clear you are talking about early magmatic minerals, which excludes quartz by definition.

We would prefer to keep this specification since it is likely that olivine was always present during the early-magmatic stage, and was subsequently resorbed after crustal assimilation only in those samples that reached quartz crystallization during the late-magmatic stage.

L. 106: This is a very interesting and important result of your study, in my opinion.

We thank the Reviewer for the kind appreciation.

L. 111: It is interesting to see how much water would have been released. What kind of impact would that have on the environment, if any?

Even if also H₂O is a greenhouse gas, we did not focus on H₂O release since it would have been much less impacting compared to CH₄ release, and its impact would be rapidly mitigated by the hydrological cycle.

L. 130: Roughly how many km wide would the aureoles have been? Can you provide a range here to help give the reader a feel for the dimensions involved.

We would prefer to express the thickness of metamorphic aureoles as percentage of the thickness of sills, since their thickness depends not only on the thickness of the intruded sills, but also on the spacing

between them (i.e., if the spacing between sills is similar to their thickness, the effective thickness of metamorphic aureoles is strongly reduced). On the other hand, repeated sill intrusions heat up host rocks progressively (cf. Annen et al., J. Petrol., 2006). Geophysical data and borehole investigations do not provide such detail to reconstruct the thickness of metamorphic aureoles within the whole Amazonas and Solimões Basins. Starting from 1 km as maximum cumulative thickness of sills in both basins, we could roughly estimate up to 5 km as potential maximum thickness of metamorphic aureoles. However, this value would approach or even exceed the maximum depth of these basins at the end-Triassic.

L. 141-143: I am not fully convinced by this statement on lines 141 to 143...you have already shown that a lot of assimilation was required to form quartz, so I do not see why assimilation of shale could not have contributed (even if in a minor way) to the CH₄ budget of the CAMP. I would suggest to "soften" this statement a little and leave the possibility open for shale assimilation as a contributor of methane. These process are not yet fully understood so it might a little premature to exclude it completely at this point.

Good point. We ruled out a direct magmatic assimilation of organic matter to form CH₄ without excluding that organic matter from assimilated shales mostly contributed to the CH₄ budget of the CAMP. We proposed that potential CH₄ generation from organic matter within intruded sedimentary rocks occurred in the metamorphic aureoles, prior to CH₄ migration into the magmatic system. At increasing temperature due to sill intrusions, it is likely that devolatilization reactions occurred prior to melting of the host sedimentary rocks, leading to decoupling of volatiles and the anatectic melt. This is indicated by the different arrows in Figure 6a. Moreover, the magmatic temperatures would allow the stability of CH₄, but would not allow its formation.

L. 155-159: I like the inclusion of a modern-day analogue here.

Many thanks. We think it could help the reader to better understand the phenomenon, even if on a different scale.

L. 162: Magma-sediment interaction (instead of magma-methane interaction)?

Magma has to interact with or heat up the sediments to mobilise methane, right? So you are looking at magma-sediment interaction (to my mind, but feel free to clarify in the way you see fit).

Good point. We agree and appreciate this important point raised by the Reviewer. Therefore, we decided to clarify this point, changing "magma-methane interaction" into "magma-sediment interaction" in the title and throughout the Main Text and the Supplementary Information.

Figure 1, panel a: Consider to label cpx too.

Added as suggested.

Figure 2, panels b, c: The bright green type is very difficult to read on my screen. Consider changing to a darker green or put a thin black outline on the letters. (Especially as you want the reader to see that this is gaseous CH₄ right away!)

Changed with thin black outline for both green and blue types.

Figure 3, panel a: Very convincing data!

Figure 3, panel b: What exactly am I looking at here? Perhaps indicate the crystal core and/or rim to help the reader get oriented.

Changed with Greek letters indicating the transect, as specified in the caption.

Figure 3, caption: Does the red profile in (b) correspond to either of the lines in panels (c) and (d)? This could be clarified a little.

See earlier point.

Figure 4: Is it possible to make this figure slightly larger? It is really a useful and well made figure so I'd love to see it a little better!

The crystallisation sketches below are also excellent. Where are the sulfides by the way? Do they come into the picture at all?

We provide the vectorial file for each figure in order to set the final size of each figure according to the final layout of the Main Text. We agree that it would be better to make it slightly larger.

In the crystallization sketches, we represented only the most abundant minerals for each crystallization stage. Sulfides occurred in subordinate amounts during the late-magmatic stage (pyrite) and during the hydrothermal stage (chalcopyrite and pyrrhotite), as described in the Supplementary Information.

L. 209: ..."which were emplaced..."

Changed as “emplaced” and moved to the Main Text.

Reviewer #2 (Remarks to the Author):

Dear authors,

It was a pleasure reading your paper, which is well written, well structured and nicely illustrated. The paper is certainly timely and will surely attract a wide spectrum of readers. It is relevant for all research carried out on LIPs and their impact on palaeoclimate and biota.

I have very few comments and suggestions, which you can find below.

In general, I think your paper could benefit from a few sentences explaining why the methane found in the fluid inclusions is not a just part of the magmatic process. I know that you have stated this in the paper, but not all readers who will be interested in this paper will have insights into magma and contact metamorphic processes. But this is just a suggestion.

The Figures are nice. The supplementary tables are well organized and supplementary information also.

I will recommend minor revision.

Good luck with the review!

Best regards

Sofie Lindström

Specific comments

Line 27. Replace “is” before synchronous with “was”.

Changed as suggested.

Line 28–30. Wouldn't it be better to write: “Here we show direct evidence that the interaction between basaltic magmas and the host sedimentary sequence produced methane during the emplacement of this large igneous province”.

Changed as “Here, we show direct evidence for the presence in basaltic magmas of methane, generated or remobilized from the host sedimentary sequence during the emplacement of this Large Igneous Province”.

Line 40. Maybe remove “the” before “Large Igneous Provinces” or replace it with “two”.

Changed as suggested.

Line 45–46. Why not “atmospheric system”?

We would prefer to keep the term “surface system” in order to include both hydrosphere and atmosphere, implying also potential interactions with the biosphere.

Line 47–50. I think that here you should mention that thermogenic C release has been hypothesized and modelled by Heimdal et al. 2020. That paper is already cited in your manuscript, but it could be mentioned here.

We agree and added this consideration, citing Heimdal *et al.* (2018; 2020).

Line 90–91. Where does the halite come from? I can't find any information about this in the supplement either. Do you expect the halite and the CH₄ to have been generated by the mixture with the same sedimentary rocks?

The origin of fluids in the H₂O-NaCl-CH₄ system (representing the content of investigated FIs) is described mainly using the constraints on CH₄, which is the most characterizing component of this system. These fluids were released during heating of the intruded Paleozoic sedimentary sequence. In more detail, NaCl may partially have a juvenile magmatic origin, and mostly derive from the intruded sedimentary rocks by assimilation of connate brines or evaporites. In order to clarify this point, some information has been moved from the Supplementary Information to the Main Text, in the Methods (subheaded section “Quantification of H₂O, NaCl and CH₄”).

Line 107–108. It would be good with a reference to back up the last part of the sentence.

Changed as suggested.

Line 108–110.

Line 111–113. This “globally fluxed” seems premature here. Perhaps better to focus on the observations first?

We agree and removed this interpretation at this point.

Line 133–136. Has it been proven to have been present already at 201 Ma, or is this a hypothesis?

It has been proven by previous studies on hydrocarbons in these Brazilian basins. In order to clarify this point, the information about the hydrocarbon system, along with relative references, has been moved from the Supplementary Information to the Main Text, at the beginning of the Results and in the caption of Figure 1.

Line 143. Maybe instead: “CH₄-bearing fluids must have migrated into ... from source or reservoir rocks”

We would prefer to keep this order of the sentence (i.e., specify the origin of the fluids before their destination), in order to leave the following “with records of their presence entrapped by FIs” close to “gabbroic sills”, where FIs are actually preserved.

Line 165. I don’t like the “surface system” term. Everything above to a certain height the surface is part of the atmosphere.

As explained for a previous comment, we would prefer to use the term “surface system” in order to include both hydrosphere and atmosphere.

Line 173–175. I think here you should stress that it is not merely the LIP magmatism but its interactions with rocks, fluids and gases in sedimentary basins, that determines the impact the LIP volcanism will have on the geobiosphere.

Changed as suggested.

Fig. 1. The photos are rather small. Otherwise fine.

We agree that it would be better to make figures slightly larger. As explained for a previous comment by Reviewer #1, we provide the vectorial file for each figure in order to set the final size of each figure according to the final layout of the Main Text.

Line 321. State why the 20 samples were chosen. Preservation?

Changed as “Mainly based on the variability in mineralogical composition and hydrothermal alteration...”.

Reviewer #3 (Remarks to the Author):

This is a well written study into the petrological evidence for basalt-sediment interactions during the Triassic–Jurassic (TJ) extinction, and the resultant production of methane and other volatiles that are proposed to have subsequently triggered the environmental and biospheric crises at that time. The approach is novel to the best of my knowledge, and the use of fluid inclusions to study these (probably thermogenic) volatile species associated with the Central Atlantic Magmatic Province could potentially be an important step forward in petrological techniques.

However, whilst the technique employed is novel, I’m not sure that the big-picture investigation is impactful enough to warrant publication in Nature Communications. Methane and other thermogenic volatiles linked to CAMP have been proposed as a major contributor to the TJ event for a long time now; the likelihood of the Brazilian basins as a source area has been commented on frequently; and the timing, quantity, and emission rates of these volatiles also modelled. So whilst obtaining petrological support for these volatile emissions is clearly an important iterative step in understanding their contribution to the TJ extinction, I don’t think it really advances our overall knowledge of the event that greatly.

Additionally, as a researcher with a limited background in igneous petrology, I found it hard to follow some of the petrological arguments at times. Of course, the length of a Nat. Comms. manuscript is prohibitive in terms of the space for expansive discussion, but given that articles on the major mass extinction events command a wide range of readers with varying interests, I think that it is important to have a little more detail in the main text, so that the arguments can be more easily followed.

In conclusion, whilst I am supportive of the study, and think that it represents a solid piece of work, I

unfortunately do not feel that it is suitable for publication in Nature Communications. I think that it could work well in a slightly longer format journal such as PNAS or EPSL though.

Minor comments:

Abstract: I wouldn't expect there to be citations in the abstract.

The references have been removed from the abstract, according to the journal guidelines.

L. 24: 'coincide' should be 'coincided'.

Changed as suggested.

L. 26: Change 'changes triggering biotic crises' to 'changes that triggered the biotic crises'.

Changed as "changes, which triggered the biotic crises".

L. 27: Strictly speaking, the CAMP isn't (or wasn't) an event, it is a geological phenomenon. The voluminous emplacement basalts to form the CAMP was the event.

Changed as "One of Earth's largest magmatic events is represented by the Central Atlantic Magmatic Province...".

L. 29: Change 'for' to 'of'. Also, this statement is slightly misleading. The host sedimentary sequences being referred to here are in northern Brazil, which was associated with just one part of the (vast) CAMP. I would imagine that in other areas the country lithology intruded by CAMP basalts were not nearly as volatile rich. So these sedimentary basins should not be stated as being representative of the continental crust intruded by CAMP as a whole.

Changed as "Here, we show direct evidence for the presence in basaltic magmas of methane, generated or remobilized from the host sedimentary sequence during the emplacement of this Large Igneous Province", taking into account also the comments by Reviewers #1 and #2.

L. 33: '...unveil the deadly instigator of the global climate change that led to the end-Triassic mass extinction.' There are a couple of things that I don't like with the ending of this sentence. Firstly, I wouldn't

say that this study unveils this methane source, as one could argue that it was first unveiled by the well-documented $\delta^{13}\text{C}$ excursions. Secondly, whilst the link between thermogenic volatiles and the extinction is increasingly compelling, I would still suggest that this is a little bit too bold, as we don't yet fully understand to what extent the climate change was driven by thermogenic vs magmatic vs other carbon sources.

Good point. Apart from the intrinsic aim to hook the reader in through the last sentence of the abstract, we would prefer not to change it since the direct geological evidence for thermogenic carbon at the end-Triassic (i.e., preserved CH_4 in CAMP basaltic rocks) was lacking before our study. Even if such phenomena were hypothesized and modelled for this and other LIPs, we think that our CH_4 -bearing FIs unveil the origin of those well-documented $\delta^{13}\text{C}$ excursions, which would be otherwise arbitrarily correlated to destabilization of methane clathrates or other negative $\delta^{13}\text{C}$ sources. Lastly, we agree that we don't yet fully understand to what extent the climate change was driven by thermogenic/magmatic/other carbon sources. However, we stated that the CH_4 preserved by our FIs contributed to the global climate change that led to the end-Triassic mass extinction, as trigger/driver of this mechanism without excluding the occurrence of any other potential carbon source.

L. 38–41: LIP volcanism also coincided with the KPg, and arguably Late Devonian, extinctions. But the link, if any, between volcanism and those two extinctions (and the end-Ordovician) remains unclear, so it's hard to confidently state that all extinctions were caused by LIPs. And lots of LIPs didn't coincide with extinctions. So what point are the authors making here?

We agree that many aspects of the causal relationship between mass extinctions and LIPs still have to be clarified, but we only reported the time coincidence between the largest mass extinctions and some LIPs. Furthermore, we agree that not all LIPs coincide in time with extinction events. However, it is not the aim of our study to discuss and focus on this point.

L. 42–45: This probably isn't true of all LIPs: at least some of them don't seem to have intruded volatile-rich sedimentary basins (Paraná-Etgedeka LIP, for example, and indeed all of the oceanic LIPs).

Good point. In the previous sentence, we specified that LIPs "often" (i.e., not always) intruded carbon-rich sedimentary basins.

L. 65: 'basin' should be 'Basin'.

Changed as suggested, throughout the Main Text and the Supplementary Information.

L. 97–98: Something feels wrong to me about giving the temperature ranges with the higher number first. Is this normal in petrology research? If not, I'd switch them to 600-700 and 400-500.

We used this conventional rule of Igneous Petrology since the described processes occurred during temperature decrease, from “hot” magma to “cold” solidified.

L. 111: What sort of degassing efficiency is assumed in order to generate this volume of emitted H₂O? 100%? That feels unrealistic though.

We stated that this was the amount of H₂O released by the entire volume of Amazonian CAMP sills (i.e., not necessarily entirely emitted into the atmosphere-hydrosphere system). Similarly to CH₄, a minimal part of H₂O was entrapped within FIs and the majority of it was either reintroduced into the host sedimentary sequence or discharged into the surface system, as explained in the Discussion.

L. 129–137: Presumably the generation temperature of thermogenic methane depends on both the depth at which this occurring, and the thermal maturity of the sedimentary rocks prior to intrusion. To what extent is this known? And are there any quantitative constraints on how much methane existed prior to CAMP versus thermogenic production during the LIP emplacement. I know that this is talked about in the supplementary text, but it is quite an important issue that I'd rather see in the main text if possible (another reason that a slightly longer-format manuscript might work better).

We fully agree about this point. In order to clarify this point, most of the information about hydrocarbon system has been moved from the Supplementary Information to the Main Text, at the beginning of the Results and in the caption of Figure 1.

L. 144–145: The conventional mechanism by which thermogenic volatiles reach the atmosphere is via breccia pipes formed due to the buildup of pressure by the newly generated volatiles. But here the authors are referring to methane escaping to the atmosphere having first been in the magma. So, are they proposing that magmatic release of thermogenic volatiles was the major mechanism of their emission instead of breccia pipe explosions? Or a supplementary route? And if the latter, do they have any ideas regarding what percentage of thermogenic volatiles were emitted via magma eruptions versus breccia pipes?

We quantified the amounts of H₂O, NaCl and CH₄ that globally fluxed through the Amazonian CAMP sills during late-magmatic quartz crystallization. As explained for a previous comment, most of these volatiles were either reintroduced into the host sedimentary sequence or discharged into the atmosphere-hydrosphere system. And the latter process likely occurred through hydrothermal vents and pipes. Since the investigated processes imply shallow intrusions only and there is no evidence for magma eruptions in the Amazonian basins, we did not discuss any emission mechanism alternative to hydrothermal vents and pipes.

L 154: change 'is' to 'has been'.

Changed as "...and was thermodynamically modelled..."

L. 156–159: This is an interesting modern analogue, which I wasn't aware of previously. I'm curious as to why the authors also don't mention Vesuvius and/or Etna here? OK, I know that they erupt through carbonates rather than muds, so there isn't a directly analogous production of thermogenic methane, but it's still another well-known modern example of thermogenic volatiles increasing the carbon output of a volcanic system following magma-sediment interaction.

Good point. As explained for a previous comment, there is no evidence for magma eruptions in the Amazonian basins. For this reason, we discussed a modern analogue of shallow magmatic intrusions within a sedimentary sequence, even if associated with a volcanic complex, and did not consider volcanoes, such as Vesuvius and Etna, which would imply magma eruptions.

L. 166–167: See earlier point. How much thermogenic carbon was released to the atmosphere via pipe systems vs through magmatic degassing following assimilation of the sedimentary rock into the magmas?

See earlier point.

Reviewers' Comments:

Reviewer #1:

Remarks to the Author:

Second review of "Massive methane fluxing from magma-sediment interaction in the end-Triassic Central Atlantic Magmatic Province" by Manfredo Capriolo et al.

After reading the revised form of the manuscript, I am satisfied that the authors have given my comments due consideration. They have implemented most of my suggestions and in a few cases where they have a different opinion, they clearly explain their thinking. The manuscript reads well with clear conclusions and the revised figures are looking good. I have no further comments. This paper will likely be of great interest to the scientific community and I look forward to seeing it published.

Reviewer #3:

Remarks to the Author:

The authors have done a good job of addressing most of my previous concerns with the manuscript, and I found the article much easier to follow this time around. I am happy that they have moved some extra details from the supplementary text into the main manuscript, which definitely support their case. I also agree with reviewer's 1 recommendation of the changed title, and much prefer the new one.

I'm still not wholly convinced that the message of there being lots of CH₄ release from the CAMP that helped trigger the TJ mass extinction is necessarily high-impact enough for Nat Comms, but I do agree that the paper presents a very interesting and novel way to strengthen this previously posited hypothesis, and I would now support publication of the article in this journal.

My recommendation to resolve this issue is to tweak the final sentence of the abstract, which I argued for before, and is the only major revision that I still have. I appreciate the authors' rebuttal to my previous comment, and their reasons for why they didn't want to change it, but the key point of this study is (to use their words) that they provide the first direct geological evidence for thermogenic carbon, not to show (or unveil) such a carbon source as the likely a driver of the extinction (which has been posited by several previous works). So I think they can change the sentence to both more accurately describe what they are actually showing, and the novelty of this approach, whilst still hooking the reader.

Something like:

"These micrometer-sized imperfections in quartz crystals provide the first demonstration that magma-sediment interactions generated huge thermogenic carbon outputs that helped instigate the global climate changes responsible for the end-Triassic mass extinction."

Otherwise, this is looking like a good piece of work that will advance the field and hopefully inspire similar studies on other LIPs. If my one comment above can be resolved, I would recommend the manuscript be accepted. Well done!

Lawrence Percival

Other (minor) comments:

Penultimate sentence on page 2:

In the case of the TJ, carbon probably is the big factor. But in theory, other volatile species could also be at work. So maybe e.g., would be better than i.e., to highlight the nuance that whilst carbon is key, it is just one example of a potentially climate-forcing gas.

Penultimate sentence on page 3:

'with the intrusion of CAMP sills'. Be specific, as I assume it is the CAMP sills (rather than any other

intrusive magmatism) that are being hypothesized as causing this thermal anomaly to generate/remobilize hydrocarbons.

End of second sentence on page 4:

'in THESE hydrocarbon systems'. Again, it is important to be specific. The CAMP did not impact all hydrocarbon systems. Picky, I know. But important.

First sentence of final paragraph of discussion:

The magma-sediment interaction is also recorded by the mineral fraction of the sills. As the authors described above, the presence of quartz in some of these sills already demands interaction with (and some degree of assimilation of) silica-rich sedimentary lithologies.

Capriolo et al., "Massive methane fluxing from magma-sediment interaction in the end-Triassic Central Atlantic Magmatic Province"

Response to further Reviewers' comments.

All comments are reproduced below and our responses follow in bold text. Corrections are shown as tracked changes in the manuscript.

We would like to reiterate our thanks to all Reviewers for their effort in improving our manuscript.

Reviewer #1 (Remarks to the Author):

Second review of "Massive methane fluxing from magma-sediment interaction in the end-Triassic Central Atlantic Magmatic Province" by Manfredo Capriolo et al.

After reading the revised form of the manuscript, I am satisfied that the authors have given my comments due consideration. They have implemented most of my suggestions and in a few cases where they have a different opinion, they clearly explain their thinking. The manuscript reads well with clear conclusions and the revised figures are looking good. I have no further comments. This paper will likely be of great interest to the scientific community and I look forward to seeing it published.

We thank the Reviewer for the constructive and positive review.

Reviewer #3 (Remarks to the Author):

The authors have done a good job of addressing most of my previous concerns with the manuscript, and I found the article much easier to follow this time around. I am happy that they have moved some extra details from the supplementary text into the main manuscript, which definitely support their case. I also agree with reviewer's 1 recommendation of the changed title, and much prefer the new one.

I'm still not wholly convinced that the message of there being lots of CH₄ release from the CAMP that helped trigger the TJ mass extinction is necessarily high-impact enough for Nat Comms, but I do agree that the paper presents a very interesting and novel way to strengthen this previously posited hypothesis, and I would now support publication of the article in this journal.

My recommendation to resolve this issue is to tweak the final sentence of the abstract, which I argued for before, and is the only major revision that I still have. I appreciate the authors' rebuttal to my previous

comment, and their reasons for why they didn't want to change it, but the key point of this study is (to use their words) that they provide the first direct geological evidence for thermogenic carbon, not to show (or unveil) such a carbon source as the likely a driver of the extinction (which has been posited by several previous works). So I think they can change the sentence to both more accurately describe what they are actually showing, and the novelty of this approach, whilst still hooking the reader.

Something like:

“These micrometer-sized imperfections in quartz crystals provide the first demonstration that magma-sediment interactions generated huge thermogenic carbon outputs that helped instigate the global climate changes responsible for the end-Triassic mass extinction.”

Otherwise, this is looking like a good piece of work that will advance the field and hopefully inspire similar studies on other LIPs. If my one comment above can be resolved, I would recommend the manuscript be accepted. Well done!

Lawrence Percival

We thank the Reviewer for the constructive and positive review, and we accept the recommendation about the final sentence of the abstract, changed as “These micrometre-sized imperfections in quartz crystals attest an extensive release of methane from magma-sediment interaction, which likely contributed to the global climate changes responsible for the end-Triassic mass extinction”. Also in order to not exceed the limit of 150 words for the abstract.

Other (minor) comments:

Penultimate sentence on page 2:

In the case of the TJ, carbon probably is the big factor. But in theory, other volatile species could also be at work. So maybe e.g., would be better than i.e., to highlight the nuance that whilst carbon is key, it is just one example of a potentially climate-forcing gas.

Changed as suggested.

Penultimate sentence on page 3:

'with the intrusion of CAMP sills'. Be specific, as I assume it is the CAMP sills (rather than any other intrusive magmatism) that are being hypothesized as causing this thermal anomaly to generate/remobilize hydrocarbons.

We would prefer not to change it since this sentence begins with the specification "During CAMP emplacement". Moreover, the following sentence clarifies that the thermal anomaly is induced by the CAMP magmatic intrusions.

End of second sentence on page 4:

'in THESE hydrocarbon systems'. Again, it is important to be specific. The CAMP did not impact all hydrocarbon systems. Picky, I know. But important.

Changed as suggested.

First sentence of final paragraph of discussion:

The magma-sediment interaction is also recorded by the mineral fraction of the sills. As the authors described above, the presence of quartz in some of these sills already demands interaction with (and some degree of assimilation of) silica-rich sedimentary lithologies.

We agree, but at this point we would prefer not to add any consideration about the mineralogical composition of the investigated CAMP sills since this sentence focuses on the methane fluxing, which is constrained by the FIs and not by the mineralogy or petrology of the basaltic rocks.